# Molecular and Functional Characterization of Pheromone Binding Protein 2 from *Cyrtotrachelus buqueti* (Coleoptera: Curculionidae)

**DOI:** 10.3390/ijms242316925

**Published:** 2023-11-29

**Authors:** Long Liu, Fan Wang, Wei Yang, Hua Yang, Qiong Huang, Chunlin Yang, Wenkai Hui

**Affiliations:** National Forestry and Grassland Administration Key Laboratory of Forest Resources Conservation and Ecological Safety on the Upper Reaches of the Yangtze River, College of Forestry, Sichuan Agricultural University, Chengdu 611130, China; liulong@sicau.edu.cn (L.L.); 2019204010@stu.sicau.edu.cn (F.W.); 10625@sicau.edu.cn (W.Y.); 12062@sicau.edu.cn (Q.H.); 14499@sicau.edu.cn (C.Y.); 14446@sicau.edu.cn (W.H.)

**Keywords:** *Cyrtotrachelus buqueti*, pheromone-binding proteins, binding assay, molecular docking, RNA interference, electroantennography

## Abstract

Pheromone-binding proteins (PBPs) play important roles in binding and transporting sex pheromones. However, the *PBP* genes identified in coleopteran insects and their information sensing mechanism are largely unknown. *Cyrtotrachelus buqueti* (Coleoptera: Curculionidae) is a major insect pest of bamboo plantations. In this study, a novel *PBP* gene, *CbuqPBP2*, from *C. buqueti* was functionally characterized. *CbuqPBP2* was more abundantly expressed in the antennae of both sexes than other body parts, and its expression level was significantly male-biased. Fluorescence competitive binding assays showed that CbuqPBP2 exhibited the strongest binding affinity to dibutyl phthalate (*K*_i_ = 6.32 μM), followed by styrene (*K*_i_ = 11.37 μM), among twelve *C. buqueti* volatiles. CbuqPBP2, on the other hand, showed high binding affinity to linalool (*K*_i_ = 10.55), the main volatile of host plant *Neosinocalamus affinis*. Furthermore, molecular docking also demonstrated the strong binding ability of CbuqPBP2 to dibutyl phthalate, styrene, and linalool, with binding energy values of −5.7, −6.6, and −6.0 kcal/mol, respectively, and hydrophobic interactions were the prevailing forces. The knockdown of *CbuqPBP2* expression via RNA interference significantly reduced the electroantennography (EAG) responses of male adults to dibutyl phthalate and styrene. In conclusion, these results will be conducive to understanding the olfactory mechanisms of *C. buqueti* and promoting the development of novel strategies for controlling this insect pest.

## 1. Introduction

Since insecticides emerged, the main strategy for fighting insect pests has been via chemical control. However, the indiscriminate use of chemical pesticides seriously harms the environment and living beings [1]. The volatile chemicals already present in the environment have been extensively studied to investigate their potential role in manipulating insect behaviors, and techniques based on these behaviors are applied to integrated pest management [2]. The chemical perception of volatile chemicals in insects depends on the olfactory system, which enables insects to achieve important tasks, such as searching for food, mating, host finding, finding a place to oviposit, and even predator avoidance [3].

Insect olfaction relies on small, specialized organs called sensilla, which are located in antennae [4]. Sensilla are exposed to chemical stimuli, and the neuronal dendrites inside are protected. The sensillar lymph is between the cuticle and the dendrites [5]. As the first filter of olfactory information for insects, small soluble extracellular proteins in the lymph, namely odorant-binding proteins (OBPs) and chemosensory proteins (CSPs), transport odorant molecules to the odorant receptors (ORs) located in olfactory receptor neurons (ORNs) [6,7]. There are different types of transmembrane proteins in ORNs, such as ORs, gustatory receptors (GRs), and sensory neuron membrane proteins (SNMPs) [8]. Receptors (e.g., ORs and GRs) enable signal transduction followed by a cascade of events to unleash behavioral responses [9].

The first insect OBP was identified in the giant moth *Antheraea polyphemus* [10]. Since then, a growing number of OBP homologues have been discovered from various insect species of different orders [11,12,13,14] and mainly from Lepidoptera [7]. Based on their similarities in amino acid sequences and functional characteristics, Lepidoptera OBPs have been divided into three subfamilies: general odorant-binding proteins (GOBPs), pheromone-binding proteins (PBPs), and antennal-binding protein (ABPx) or antennal-specific proteins (ASPs) [10,15,16].

Insect PBPs are soluble proteins with six conserved cysteine residues, and they are mainly localized in the sensillum lymph of antennae and male-biased in expression, playing important roles in binding and transporting the sex pheromone substances produced by females [10,17]. Except for binding to sex pheromones, some studies have indicated that PBPs can also bind to plant volatiles [18]. So far, although PPBs were also identified and characterized in coleopteran insects such as *Anoplophora glabripennis* [19], *Ophraella communa* [20] and *Holotrichia oblita* [21], the *PBP* genes identified in Coleoptera and their information-sensing mechanism are still poorly known. Thus, a lot of work is required to investigate and better understand the information perception of PBPs in Coleoptera insects.

The bamboo snout beetle, *Cyrtotrachelus buqueti* (Coleoptera: Curculionidae), is a major insect pest of bamboo plantations and is wildly distributed in southern China and several countries of Southeast Asia, such as Burma, Vietnam, Thailand [22]. This pest seriously damages several bamboo species, including *Neosinocalamus affinis*, *Phyllostachys pubescens*, *Dendrocalamus farinosus*, and *Bambusa textilis* [23]. This pest causes damages to bamboo shoots by piercing and sucking, as well as laying eggs [24]. Since this pest has seriously influenced the development of the bamboo industry, it has been listed as a dangerous forestry pest by the State Forestry Administration of China [25].

With the aim of developing control techniques based on the sex pheromone, 19 candidate PBPs were identified from *C. buqueti* transcriptome in our previous study [26]. In our follow-up studies, the function of PBP1 protein of *C. buqueti* (CbuqPBP1) was characterized by investigating ligand-binding properties and key binding sites [27,28]. Except for CbuqPBP1, the other PBPs are still not characterized in *C. buqueti*, and the olfactory mechanisms of *C. buqueti* are not clear. Interestingly, we found that CbuqPBP1 could interact with another PBP proteins of *C. buqueti* (CbuqPBP2) via GST pull-down and yeast two-hybrid assays (not published). Therefore, CbuqPBP2 was functionally characterized in this study to better understand the olfactory mechanism of *C. buqueti*. The expression pattern of *CbuqPBP2* regarding gender and tissue was investigated, and the CbuqPBP2 protein was successfully expressed and purified using the bacteria expression system. Then, with the purified recombinant protein, the binding properties of CbuqPBP2 to twelve *C. buqueti* volatiles and three plant volatiles from the host *N. affinis* were tested via fluorescence competitive binding assays. Moreover, molecular modeling and molecular docking were performed to understand the molecular interaction mechanisms between the CbuqPBP2 and ligands. Finally, the function of CbuqPBP2 was further confirmed in vivo using RNA interference (RNAi) followed by electroantennography (EAG) assays. This paper will be conducive to the study of the olfactory mechanism of *C. buqueti* and lay a theoretical basis for developing new pest control methods.

## 2. Results

### 2.1. Sequence Analysis of CbuqPBP2

The predicted ORF of *CbuqPBP2* was 402 bp in length, encoding 133 amino acid residues with a signal peptide of 18 amino acid residues at the N-terminus (Figure 1A). The gene sequence of *CbuqPBP2* was verified via PCR amplification and Sanger sequencing. The calculated molecular weight of CbuqPBP2 was 15.17 kDa, and the isoelectric point was 4.31. Similar to other PBPs, the CbuqPBP2 protein contained a single insect pheromone-binding protein domain (Figure 1A). As shown in Figure 1B, multiple amino acid sequence alignment revealed that CbuqPBP2 was highly similar to other coleopteran PBPs. CbuqPBP2 shared the highest identity of 53.6% with AglaPBP1 from *A. glabripennis*, followed by BhorPBP1 from *Batocera horsfieldi* (52.9% identity), while the CbuqPBP2 shared low amino acid identity (21.3%) with another PBP protein of *C. buqueti* (CbuqPBP1). The CbuqPBP2 contained six conserved cysteine residues, and the cysteine mode was C_1_-X_26_-C_2_-X_3_-C_3_-X_37_-C_4_-X_8_-C_5_-X_8_-C_6_ (X denotes any amino acid). The ORF sequence of *CbuqPBP2* was deposited in GenBank with the accession number MN400445.1.

### 2.2. Phylogenetic Analysis

As shown in Figure 2, a phylogenetic tree was constructed to gain insight into the evolutionary relationships between CbuqPBP2 and other orthologs in Lepidoptera and Coleoptera. As expected, the PBPs in Lepidoptera and Coleoptera were clustered as an independent group, respectively. *CbuqPBP2* belonged to Coleoptera branch and was placed as a sister taxon to the clade composed of AglaPBP1 from *A. glabripennis*, BhorPBP1 from *B. horsfieldi*, BpraPBP from *B. prasinus*, HparPBP1 from *Holotrichia parallela*, PjapPBP from *P. japonica*, and AcupPBP from *Anomala cuprea* with strong supports. *C. buqueti* seems to obtain the PBP proteins earlier than most other coleopteran insects.

### 2.3. Tissue and Sex Expression Pattern of CbuqPBP2

To further determine the functions of CbuqPBP2 in chemical communication, quantitative real-time PCR (qPCR) was employed to measure the expression levels of *CbuqPBP2* in different tissues of both female and male adults, including antennae, heads (without antennae), thoraxes, abdomens, and legs. The tissue and sex expression pattern of *CbuqPBP2* was similar to that of *CbuqPBP1* [27]. As shown in Figure 3, the expression of *CbuqPBP2* was significantly higher in antennae than other tissues in both male and female adults. Furthermore, the expression level of *CbuqPBP2* was significantly male-biased in the antennae. *CbuqPBP2* expression was approximately 1.69-fold higher in the male antennae than that in the female antennae.

### 2.4. Fluorescence Competitive Binding Assays

The recombinant CbuqPBP2 protein was successfully expressed in Arctic Express competent cells, and SDS-PAGE displayed a target protein band at 14.4–18.4 kDa that was consistent with the predicted molecular weight of the fusion protein (Appendix A). Using *N*-phenyl-1-naphthylamine (1-NPN) as a fluorescence probe, the purified CbuqPBP2 protein was then used in fluorescence competitive binding assays to illustrate its binding affinities with various volatiles. First, the binding of 1-NPN with CbuqPBP2 was analyzed. As shown in Figure 4A, the binding between CbuqPBP2 and 1-NPN gradually saturated as the 1-NPN concentration increased, and the Scatchard plot comprised a straight line, indicating that there is a single binding site between CbuqPBP2 and 1-NPN. CbuqPBP2 had strong binding ability with 1-NPN, and the dissociation constant (*K*_d_) of the CbuqPBP2/1-NPN complex calculated from the Scatchard plot was 6.41 μM. The above results indicated that 1-NPN can be used to analyze the binding properties of CbuqPBP2 to different ligands.

The binding abilities of CbuqPBP2 to twelve *C. buqueti* volatiles were examined, and the competitive binding curves are shown in Figure 4B,C. Among twelve volatile compounds of *C. buqueti*, CbuqPBP2 exhibited high binding affinity to dibutyl phthalate and styrene, with *K*_i_ values of 6.32 and 11.37 μM, respectively, while the binding affinity of CbuqPBP2 to ethyl hexanoate was weak, with a *K*_i_ value of 39.20 μM (Table 1). Compared to CbuqPBP2, the CbuqPBP1 also exhibited the strongest binding affinity with dibutyl phthalate but a lower binding affinity with styrene [27]. The binding affinities of CbuqPBP2 to three plant volatiles from the host *N. affinis* were also determined to further understand the role of CbuqPBP2 in finding host plants. Among three volatile compounds of *N. affinis*, CbuqPBP2 only bound with linalool and showed high binding affinity with a *K*_i_ value of 10.55 μM (Figure 4D and Table 1), while CbuqPBP1 could bind all three volatiles but with low binding affinity [27]. As shown in Figure 4C,D, 1-NPN could not displace squalene and indole. Instead, the fluorescence intensity enhanced as the ligand concentration increased, and the reasons behind this phenomenon require further investigation.

### 2.5. Structural Modeling and Molecular Docking

A 3D model of CbuqPBP2’s protein structure was constructed de novo using the AlphaFold program (Figure 5A). Our Ramachandran plot analysis indicated that 96.26% of the residues were present in the most favored regions, 2.9% were present in the additional allowed regions, 1.0% were present in the generously allowed regions, and no residue was present in disallowed regions, suggesting the accuracy of the predicted CbuqPBP2 structure (Figure 5B). The modeled CbuqPBP2 3D structure comprises six α-helices folded into a very compact and stable globular structure and six conserved cysteine residues form three pairs of disulfide bonds, which is typical for insect OBPs (Figure 5A).

A molecular docking analysis was performed using Autodock Vina 1.1.2 [29] to further elucidate the binding mode of CbuqPBP2 to dibutyl phthalate, styrene, and linalool. The strength of the binding ability of CbuqPBP2 to these three compounds was evaluated based on binding energy, with lower binding energy corresponding to a stronger binding ability. In line with the *K*_i_ values obtained from the fluorescence binding assays (Table 1), CbuqPBP2 displayed the strongest interaction with dibutyl phthalate, with a binding energy value of −5.7 kcal/mol. CbuqPBP2 also showed strong interaction with styrene and linalool, with binding energy values of −6.6 and −6.0 kcal/mol, respectively. As shown in Figure 6, there were a large number of hydrophobic amino acid residues near the CbuqPBP2-ligand complexes, indicating strong hydrophobic interactions. Dibutyl phthalate was docked in a hydrophobic cavity consisting of Leu114, Leu48, Ile115, Phe113, Ala9, Pro71, Met5, Ile12, Met52, Leu8, Val69, Met66, and His101 (Figure 6A). Styrene was docked in a hydrophobic cavity consisting of Phe113, Leu70, Tyr112, Leu78, Val81, Tyr104, Val82, and Met66 (Figure 6B). Linalool had hydrophobic interactions with Ile115, Met66, Tyr104, Tyr112, Leu78, His113, Val82, Leu70, Pro71, Val69, and His101 and a strong hydrogen-bonding interaction with Phe113 with a length of 3.1 Å (Figure 6C).

### 2.6. RNA Interference of CbuqPBP2 and Electroantennography

RNAi was employed to investigate the biological function of CbuqPBP2. Compared to water-injected and non-injected controls, *CbuqPBP2* transcription in the antennae of male adults showed a statistically significant decrease at 48, 72, and 96 h after ds*CbuqPBP2* injection (Figure 7A), indicating that *CbuqPBP2* in the antennae of male adults was successfully knocked down via RNAi.

Based on the results of our fluorescence competitive binding assays, volatiles that bound with CbuqPBP2 were selected to measure the antennal responses of male *C. buqueti* adults at 72 h after RNAi by using an EAG analysis, including three *C. buqueti* volatiles (dibutyl phthalate, ethyl hexanoate, and styrene) and one plant volatile of the host *N. affinis* (linalool). Compared with water-injected and non-injected controls, the EAG responses of ds*CbuqPBP2*-treated adults to dibutyl phthalate and styrene were significantly reduced (Figure 7B). Consistent with the result that CbuqPBP2 showed weak binding ability to ethyl hexanoate (Table 1), the EAG response of ds*CbuqPBP2*-treated adults in response to ethyl hexanoate did not significantly decrease (Figure 7B). Interestingly, the EAG response of ds*CbuqPBP2*-treated adults to linalool was not significantly reduced, though CbuqPBP2 had a relative high binding ability to linalool (Figure 7B).

## 3. Discussion

The insect olfactory system plays a crucial part in mating, oviposition, host finding, and the avoidance of natural enemies or adverse environments [10]. Serving as the initial filter, OBPs facilitate the transport of odorants via the sensillar lymph and direct them to the dendrites, triggering the activation of the target odorant receptors [7,8]. PBPs, the subfamily of OBPs, are commonly reported in lepidopteran insects and responsible for sex pheromone recognition [30]. In this study, the *CbuqPBP2* gene of *C. buqueti* was functionally characterized. The amino acid sequence of CbuqPBP2 contained six conserved cysteine residues, and the cysteine mode was in line with the motif pattern of C_1_-X_15-39_-C_2_-X_3_-C_3_-X_21-44_-C_4_-X_7-12_-C_5_-X_8_-C_6_ of typical OBPs of insects [31]. As expected, multiple sequence alignment and phylogenetic analysis showed that CbuqPBP2 was highly similar to PPBs from other coleopteran insects. However, the CbuqPBP2 shared low amino acid identity with CbuqPBP1, indicating different PBPs of *C. buqueti* may vary in their ability to recognize different odorant molecules in the environment.

The expression patterns of olfactory-related genes in various tissues and sexes can offer insights into comprehending their physiological functions [32]. The crucial involvement of antennae-enriched OBPs in detecting sex pheromones and host plant compounds has been demonstrated in numerous experiments [33,34,35]. The qPCR was used to determine the tissue and sex expression profiles of *CbuqPBP2*. The results showed that *CbuqPBP2* was significantly highly expressed in antennae of both sexes compared to other tissues, which may reflect the potential functions in olfactory chemoreception. Moreover, *CbuqPBP2* was significantly highly expressed in male antennae compared to female antennae, and similar expression patterns were also reported in many lepidopteran PBPs [36]. The tissue and sex expression pattern of *CbuqPBP2* is consistent with that of *CbupPBP1* [27], and the male-biased expression suggested that PBPs may play a crucial role for *C. buqueti* males in binding and discriminating sex pheromones released from females.

To determine the binding characteristics of CbuqPBP2, twelve *C. buqueti* volatiles and three volatiles of host plant (*N. affinis*) were selected to perform competitive fluorescence binding assays. The results revealed that CbuqPBP2 showed the highest binding affinity with dibutyl phthalate, followed by styrene, among twelve *C. buqueti* volatiles. It was demonstrated that the relative content of dibutyl phthalate in female *C. buqueti* was higher than that in males [37] and that ester compounds are key sex pheromone components in some coleopteran insects [38,39]. The PBP1 and PBP2 of *Agrotis ypsilon* exhibited selective binding to sex pheromone components, but both of them bound with major components [40]. Similarly, the binding affinity of CbuqPBP2 to twelve *C. buqueti* volatiles was obviously different from the PBP1 protein of *C. buqueti* [27], but both showed the strongest binding affinity with dibutyl phthalate. Compared with mated males and virgin females, dibutyl phthalate showed a more significant attraction effect on the virgin males of *C. buqueti* [41]. Therefore, it was speculated that dibutyl phthalate may be the primary component of the *C. buqueti* sex pheromone. Although CbuqPBP2 also showed high binding affinity with styrene, further behavioral evidence is required to determine the possibility of styrene as a sex pheromone component of *C. buqueti*. Plant volatile compounds serve as the main olfactory signals utilized by herbivorous insects to locate and navigate towards food sources and oviposition sites [42]. In addition to binding sex pheromones, recent studies have revealed that PBPs also show binding affinities to host plant volatiles, such as the PBPs of the yellow peach moth *Conogethes punctiferalis* [43], suggesting that insect PBPs have a dual function of recognizing sex pheromones and host-derived volatiles. However, little is known about the function of coleopteran PBPs in recognizing host plant volatiles. In our previous study, it was found that CbuqPBP1 could bind to all three host-derived volatiles tested in this study, but the binding affinity was weak [27]. In contrast to CbuqPBP1, CbuqPBP2 only bound to linalool and exhibited high binding affinity, indicating that *C. buqueti* PBPs selectively bind host plant volatiles.

In order to further understand the molecular interaction mechanisms, the binding mode of CbuqPBP2 to dibutyl phthalate, styrene, and linalool was determined via molecular docking analysis. In molecular docking, the binding strength was assessed by the alteration in potential energy around the binding pocket during the interaction between the protein and ligand [44]. Consistent with the fluorescence binding assays, the docking results showed lower binding energies between CbuqPBP2 with three tested compounds, suggesting strong binding strength. It was proved that PBPs commonly bind ligands in hydrophobic cavities, such as PBPs from *Orthaga achatina* [45]. In our study, it was also found that hydrophobic interactions were the prevailing forces within the binding cavities of CbuqPBP2. Furthermore, hydrogen bonding is also involved in the interaction between CbuqPBP2 and linalool, which was able to promote binding, indicating that there are different binding mechanisms for different types of volatile compounds.

A combination of RNAi and EAG was used to confirm the function of CbuqPBP2 in *C. buqueti* olfaction. The gene expression level of *CbuqPBP2* was successfully knocked down, and the EAG responses to certain compounds were affected by the reduced level of *CbuqPBP2* transcripts. Consistent with the results of our competitive fluorescence binding assays, the EAG responses of *CbuqPBP2*-dsRNA-treated males to dibutyl phthalate and styrene were significantly reduced compared with non-injected and water-injected controls. These results provide further evidence that CbuqPBP2 plays an important role in detecting and transporting these two volatiles. However, the transcription level of *CbuqPBP2* did not influence the EAG response to linalool. The possible reason for this is that the recognition of a specific compound may involve the participation of multiple OBPs. For instance, the OPB2 and OBP3 of *Cnaphalocrocis medinalis* may work synergistically to mediate the recognition of odorants [46].

## 4. Materials and Methods

### 4.1. Sequence and Phylogenetic Analysis

The PBP gene of *C. buqueti* (*CbuqPBP2*) was obtained by searching a previously annotated transcriptome (GenBank accession number: SRS1876730) [26]. The Open reading frame (ORF) was determined using NCBI ORF finder (https://www.ncbi.nlm.nih.gov/orffinder, accessed on 17 October 2018). The SignalP-6.0 server (https://www.cbs.dtu.dk/service.php?SignalP, accessed on 7 August 2023) was applied to predict the signal peptide. The domain architecture was determined using the SMART program (http://smart.embl-heidelberg.de/, accessed on 9 August 2023). The MAFFT method was used to perform multiple sequence alignment with the auto algorithm and BLOSUM62 scoring matrix [47], and ESPript 3.0 (http://espript.ibcp.fr/ESPript/ESPript/, accessed on 20 November 2023) was used to visualize the result [48]. The molecular weight (Mw) and isoelectric point (pI) were determined using the ProtParam (Expasy) tool (https://web.expasy.org/protparam/, accessed on 13 August 2023) [49].

Phylogenetic analyses of CbupPBP2 and PBPs from other coleopteran and lepidoptera insects were performed using the maximum likelihood (ML) method. These PBPs were aligned using the MAFFT method as described above. The ModelFinder integrated in PhyloSuite (version 1.2.2) was used to determine the best-fit model of amino acid evolution for these PBPs with default settings based on the Bayesian information criterion (BIC) [50,51], and the best-fit model was LG + I + G4. Under the best-fit model estimated by ModelFinder, the IQ-TREE [52] in PhyloSuite (version 1.2.2) [51] was used to generate the ML tree with the default parameters. The branch support was estimated with 5000 ultrafast bootstraps [53], as well as the Shimodaira–Hasegawa-like approximate likelihood-ratio test (SH-aLRT) with 1000 replicates [54]. The iTOL server (https://itol.embl.de/, accessed on 15 August 2023) [55] was used to visualize the phylogenetic tree.

### 4.2. Insect Rearing and Tissue Collection

Pupae of *C. buqueti* were collected from the bamboo planting base (30°13′ E, 102°91′ N) in Lushan County, Ya’an City, Sichuan Province, China. Collected pupae were reared in our laboratory under the conditions of 25 ± 1 °C, 70% ± 10% relative humidity, and a 12 h light:12 h dark photoperiod. After emergence, adults were sexed and reared on fresh *N. affinis* shoots in insect cages (60 × 60 × 60 cm) under the same conditions. For tissue-specific gene expression, the heads (without antennae), antennae, thoraxes, abdomens, and legs were dissected from male and female *C. buqueti* adults, respectively. One sample contained at least 5 individual tissues, and more than five sample replicates were prepared. All samples were flash-frozen in liquid nitrogen and then preserved at −80 °C until further use.

### 4.3. Total RNA Extraction, cDNA Synthesis, and qPCR

Total RNA was isolated using the MiniBEST Universal RNA Extraction Kit (TaKaRa, Dalian, China) following the manufacturer’s protocol. The RNA integrity was determined via 1% agarose gel electrophoresis. RNA concentration and quality (OD260/280 > 1.8) were determined using the DU800 spectrophotometer (Beckman Coulter, CA, USA). According to the manufacturer’s instructions, first-strand cDNA was synthesized from 1 μg of total RNA using the PrimerScript^TM^ RT Reagent Kit (Perfect Real Time, TaKaRa, Dalian, China). The final cDNA samples were stored at −20 °C until further analysis.

The tissue-specific expression of *CbPBP2* was determined via quantitative real-time PCR (qPCR). cDNA was diluted 1:5 in sterilized PCR-grade water, and qPCR was run on an CFX96™ Real-time PCR detection system (Bio-Rad, CA, USA) using the TB Green™ Premix Ex Taq™ II (Tli RNaseH Plus) Kit (TaKaRa, Dalian, China). The volume of each reaction was 25 µL, containing 12.5 µL of TB Green Premix Ex Taq II (Tli RNaseH Plus), 2 µL of DNA template, 1 µL each of sense and antisense primers (10 µM), and 8.5 µL of sterilized PCR-grade water. All samples were analyzed in triplicate under the following reaction program: an initial denaturation for 30 s at 95 °C, followed by 40 cycles of 95 °C for 5 s, and 60 °C for 30 s. The specificities of amplified products were assessed via a melting curve analysis. Gene-specific primers (Appendix A) were designed using the primer premier software (version 5.0). The Glyceraldehyde 3-phosphate dehydrogenase (*GAPDH*) gene (GenBank accession number: KY745870.1) was used as the qPCR internal control gene. A standard curve was derived from a 10-fold serial dilution of the plasmid containing the target DNA segment to determine the PCR efficiency of each primer pair for the target and reference genes. All the primers produced amplification efficiencies of 95% to 105%. The relative mRNA expression levels of *CbuqPBP2* were determined using the 2^−ΔΔCt^ method [56].

### 4.4. Preparation of Recombinant CbuqPBP2 Protein

Using the CE design tool (https://crm.vazyme.com/cetool/multipoint.html, accessed on 18 March 2019), gene-specific primers (Appendix A) were designed to amplify the cDNA encoding the mature CbuqPBP2 protein from the male antennal cDNA. According to the manufacturer’s instructions, the products of the polymerase chain reaction (PCR) were cloned into a pET-28a (+) bacterial expression vector using the ClonExpress II One Step Cloning Kit (Vazyme, Nanjing, China) and verified via sequencing (entrusted to Sangon Biotech Co., Ltd., Shanghai, China). The recombinant plasmid was transformed into competent cells of the *Escherichia coli* Arctic Express (DE3) expression strain. The verified bacterial suspension was inoculated into liquid LB medium containing 50 μg/mL kanamycin and incubated at 37 °C until OD_600_ reached 0.6–0.8. The expression of recombinant CbuqPBP2 protein was induced at 37 °C for 6 h via the addition of isopropyl β-D-1-thiogalactoside (IPTG) to a final concentration of 1 mM. This optimal induction concentration was determined by inducing the expression of CbuqPBP2 with different concentrations of IPTG (Appendix A). The bacterial cells were precipitated via centrifugation (12,000 rpm, 4 °C for 10 min). The pelleted bacterial cells were resuspended in PBS (pH 7.5) and sonicated on ice. The recombinant CbuqPBP2 was purified via Ni-IDA chromatography with a gradient concentration of imidazole washing. The purified protein was dialyzed overnight in a dialysis bag against 20 mM Tris–HCl, 0.15 M NaCl, and then detected via SDS-PAGE analysis. The concentration of the purified protein was determined using Bradford Protein Assay Kit (Solarbio Science & Technology Co., Ltd., Beijing, China).

### 4.5. Fluorescence Competition Binding Assay

Previous behavioral studies employing the Y-tube olfactometer and flight tunnel have shown that adult females of *C. buqueti* were more attractive to males than males were to females. Furthermore, the volatiles emitted by females exhibited a robust attractant effect on males, indicating the likely presence of sex pheromones in females [37,57]. In addition, *C. buqueti* displayed high EAG responses to thirteen volatiles from the host plant *N. affinis* [58]. We found that CbuqPBP1 exhibited binding affinity towards twelve volatiles emitted by *C. buqueti* and three out of the thirteen volatiles released by *N. affinis* [27]. Based on the above studies, twelve volatiles from *C. buqueti* and three from *N. affinis* (Table 1) were selected to perform the fluorescence competition binding assay to explore the olfactory mechanism of *C. buqueti*.

The fluorescent probe 1-NPN and the tested chemicals with a purity of at least more than 96% were purchased from Aladdin Biochemical Technology Co., LTD. (Shanghai, China). The fluorescence binding assay was carried out on a Lumina fluorescence spectrophotometer (Thermo Fisher Scientific, MA, USA) using a 1 cm light path quartz cuvette at 25 °C. All the slit widths for both excitation and emission were 5 nm. The 1-NPN and all the compounds tested were dissolved in chromatography-grade methanol to a final concentration of 1 mM. The CbuqPBP2 protein was dissolved in 50 mM Tris–HCl buffer at pH 7.4 to a final concentration of 2 μM. To measure the affinity of 1-NPN to CbuqPBP2, a 2 μM solution of the CbuqPBP2 protein was titrated with aliquots of the 1 mM 1-NPN solution to final concentrations ranging from 2 to 30 μM. The protein/1-NPN complex was excited at 337 nm. The scanning range was 350–550 nm at each concentration, and the maximum fluorescence intensities were plotted against the free ligand concentration. GraphPad Prism 8.4.1 was used to calculate the dissociation constants (*K*_d_) of the CbuqPBP2 with 1-NPN from Scatchard plots of the binding data under the assumption that the protein was 100% active, and its stoichiometric ratio with the ligand was 1:1.

Using both 1-NPN and CbuqPBP2 at a concentration of 2 μM, the binding affinity of the odorant ligands was tested through competitive binding assays by adding each competitor ligand at different concentrations. The dissociation constants of each competitor ligand were calculated from the corresponding half-maximal inhibitory concentration (IC_50_) values using the following equation: *K*_i_ = [IC_50_]/(1 + [1-NPN]/*K*_1-NPN_), where [1-NPN] represents the free concentration of 1-NPN, and *K*_1-NPN_ represents the dissociation constant of the protein/1-NPN complex.

### 4.6. Structural Modeling and Molecular Docking

The AlphaFold2 program [59] was used to construct the three-dimensional (3D) structure of CbuqPBP2 after the signal peptide was removed. The stereo-chemical quality of the modeled 3D structure was evaluated by constructing Ramachandran plots using the pymod3 plugin implemented in PyMOL 2.4.2 (https://pymol.org/2/, accessed on 20 August 2023). Based on the florescence binding assay, compounds with an IC_50_ value of < 20 μM (Table 1) were selected for molecular docking simulations. The 3D structures of the target ligands were downloaded in SDF format from the PubChem database (https://pubchem.ncbi.nlm.nih.gov/, accessed on 20 August 2023). Autodock Vina 1.1.2 [29] with the default parameters described in the Autodock Vina manual was employed for molecular docking. The optimal docking model was selected based on the most negative binding energy, with a RMSD value of zero. PyMOL 2.4.2 was used for model visualization, and LigPlot^+^ (https://www.ebi.ac.uk/thornton-srv/software/LigPlus/, accessed on 26 August 2023) was used for drawing the 2D depictions.

### 4.7. RNA Interference of CbuqPBP2

RNAi was performed to decrease the transcription level of *CbuqPBP2* in *C. buqueti*. Double-stranded RNA (dsRNA) was synthesized using full-length male antennal cDNA of *C. buqueti* using the MEGAscript T7 Transcription Kit (Thermo Fisher Scientific, MA, USA) according to the manufacturer’s protocol. Specific primers containing a T7 polymerase promoter sequence at their 5′ end are listed in Appendix A. The integrity and concentration of the dsRNA were determined via agarose gel electrophoresis and the DU800 spectrophotometer (Beckman Coulter, CA, USA), respectively. The dsRNA was dissolved in RNase-free water at a concentration of 5 μg/μL and stored at −80 °C until further use.

The *CbuqPBP2* dsRNA (5·μg/μL × 1.5 μL) was injected into the abdomen of male *C. buqueti* adults using a microinjector. The RNase-free water-injected and non-injected adults were used as controls. After 48, 72, and 96 h, the antennae were collected. Then, qPCR was performed to assess effects of RNAi on the transcription level of the *CbuqPBP2* gene, and three replicates were performed.

### 4.8. EAG Assays after RNAi

To evaluate the function of CbuqPBP2 in *C. buqueti* olfaction, the dsRNA-injected, RNase-free water-injected, or non-injected male adults were subjected to EAG (Syntech, Hilversum, Netherlands) at 72 h after injection. Based on the results of the fluorescence competition binding assay, the chemicals (dibutyl phthalate, ethyl hexanoate, styrene and linalool) that bound with CbuqPBP2 were selected for electrophysiological experiments. The tested chemicals were diluted in n-hexane to prepare a solution at a concentration of 100 μg/μL, and n-hexane was used as a blank control.

The antenna for EAG detection was cut off and amputated at the base and tip. The basal and distal ends of the excised antenna were linked to a reference electrode and recording electrode by one teardrop of Spectra R360 electrically conductive gel, respectively. A filter paper strip (2 cm × 0.5 cm) loaded with ten microliters of each chemical solution was inserted into the sample tube (10 cm in length) as a stimuli source. An air-stimulus controller (CS-05 model) was used to deliver the stimulus. The stimulation time was 0.5 s, and the interval for antennal recovery was 30 s. The response of the antenna to each chemical was detected using a high-impedance amplifier (IDAC2) and analyzed using the EAG software (version 2.0) supplied with the instrument. The chemicals were tested randomly for each antenna. The antenna was stimulated one time with n-hexane before and after each chemical stimulation. Three to six antennae were measured for each chemical in independent experiments, and each antenna was tested three times. EAG responses were recorded in mV, and the relative EAG response of each chemical was calculated by using the following formula:$$Relative\;EAG\;response=\frac{EAG\left(A\right)}{\left({EAG}_{CK1}+{EAG}_{CK2}\right)/2}$$
where EAG(A) represents the EAG response to the chemical; EAG_CK1_ and EAK_CK2_ represent EAG responses to n-hexane before and after each chemical stimulation, respectively.

### 4.9. Statistical Analysis

GraphPad Prism 8.4.1 was applied to perform statistical analyses. Homogeneity of variance was assessed for all data prior to further analysis. Regression analyses for the binding of 1-NPN with CbuqPBP2 and the corresponding homoscedasticity test were performed using GraphPad Prism 8.4.1 software. A one-way ANOVA followed by Tukey’s multiple comparison tests were used to determine the statistical significance of the differences in the tissue-specific expression levels of the *CbuqPBP2* gene, and different lowercase letters indicate significant differences (*p* < 0.05). Student’s *t*-test with a 95% confidence interval was used to determine whether the expression levels of the *CbuqPBP2* gene and EAG values were significantly different between the *CbuqPBP2* dsRNA injection and control groups, with asterisks indicating significant differences (denoted by * *p* < 0.05 and ** *p* < 0.01). The assumption of normality was tested before Student’s *t*-test using GraphPad Prism 8.4.1.

## 5. Conclusions

In conclusion, the data obtained from the present study revealed that *CbuqPBP2* was predominantly expressed in the antennae of *C. buqueti* adults, and the expression level in antennae was significantly male-biased. Fluorescence binding assays and docking analysis indicated the dual roles of CbuqPBP2 in binding volatiles from the same species and the host plant. RNAi combined with EAG experiments further confirmed that CbuqPBP2 participated in the process of detecting and transporting two *C. buqueti* emitted by *C. buqueti* and one volatile from the host plant. The information provided in this study will lay a theoretical foundation for developing new methods to control *C. buqueti* by interfering with their olfactory perception.

## Figures and Tables

**Figure 1 ijms-24-16925-f001:**
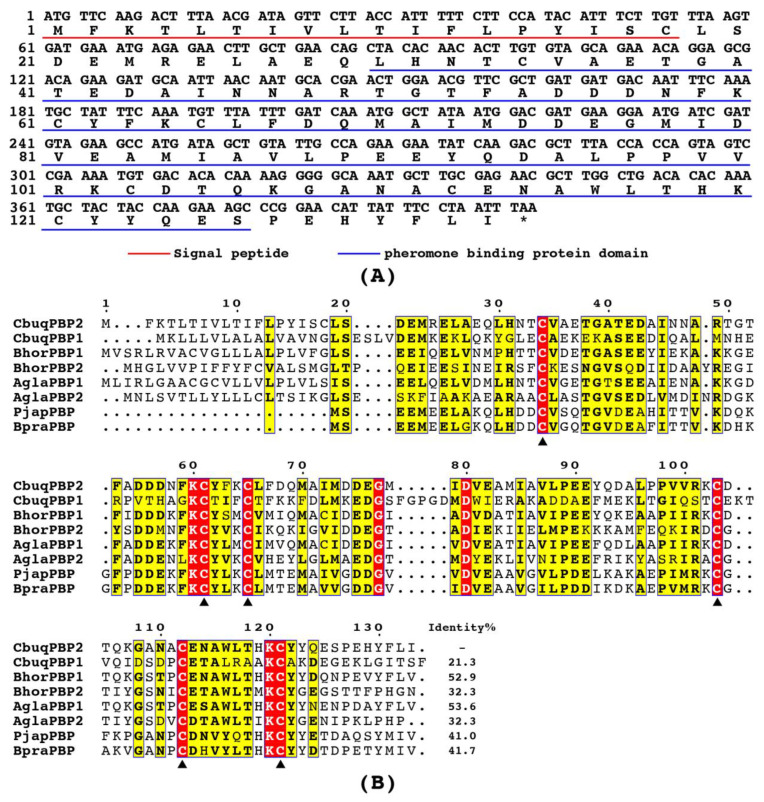
(**A**) Nucleotide and amino acid sequences of the *CbupPBP2* gene in *C. buqueti*. The signal peptide and the insect pheromone-binding protein domain are underlined in red and blue, respectively. The asterisk indicates the stop codon. (**B**) Multiple amino acid sequence alignment of CbupPBP2 with other PBPs from coleopteran insects. The species and GenBank accession numbers of these PBPs are as follows: *C. buqueti* (CbuqPBP1, AOF39987.1; CbuqPPB2, MN400445.1); *B. horsfieldi* (BhorPBP1, AIV43008.1; BhorPBP2, AIV43009.1); *A. glabripennis* (AglaPBP1, ASA46120.1; AglaPBP2, ASA46121.1); *Popillia japonica* (PjapPBP, AAC63436.1); and *Brachysternus prasinus* (BpraPBP, AGG37860.1). Six conserved cysteine residues are marked by black triangles.

**Figure 2 ijms-24-16925-f002:**
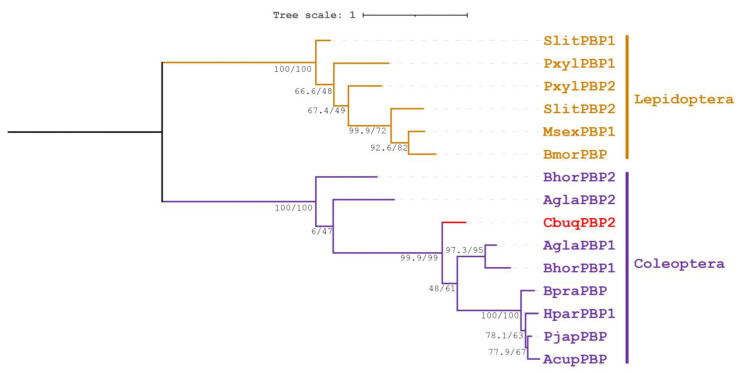
Phylogenetic analysis of CbupPBP2 and PBPs from other coleopteran and lepidoptera insects using the maximum likelihood method. Bootstrap support values and SH-aLRT values are indicated on branches. The tree is rooted by a midpoint approach. CbupPBP2 is highlighted in red. The species and GenBank accession numbers of all selected PBP sequences are as follows: *Spodoptera litura* (SlitPBP1, AKI87957.1; SlitPBP2, AKI87958.1); *Plutella xylostella* (PxylPBP1, ACI28451.1; PxylPBP2, AGH13203.1); *Manduca sexta* (MsexPBP1, AAG50016.1); *Bombyx mori* (BmorPBP, CAA64443.1); *B. horsfieldi* (BhorPBP1, AIV43008.1; BhorPBP2, AIV43009.1); *A. glabripennis* (AglaPBP1, ASA46120.1; AglaPBP2, ASA46121.1); *C. buqueti* (CbuqPPB2, MN400445.1); *B. prasinus* (BpraPBP, AGG37860.1); *H. parallela* (HparPBP1, ADF87391.1); *P. japonica* (PjapPBP, AAC63436.1); and *A. cuprea* (AcupPBP, BAC06498.1).

**Figure 3 ijms-24-16925-f003:**
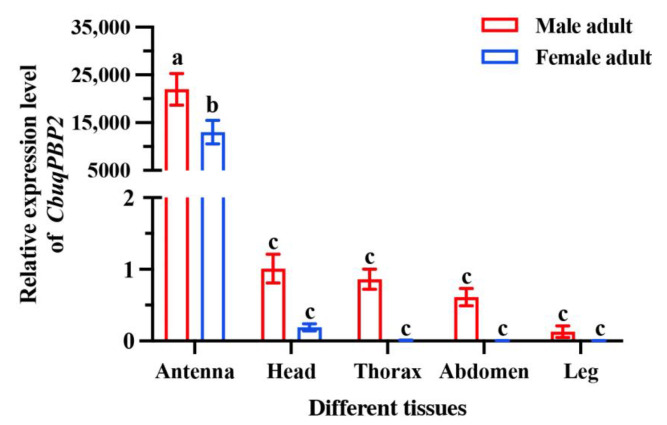
Relative expression levels of *CbuqPBP2* in different tissues of both female and male adults. Data are presented as the means ± standard deviation (SD) with three biological replicates. Different lowercase letters above the bars denote significant differences at *p* < 0.05 among different samples.

**Figure 4 ijms-24-16925-f004:**
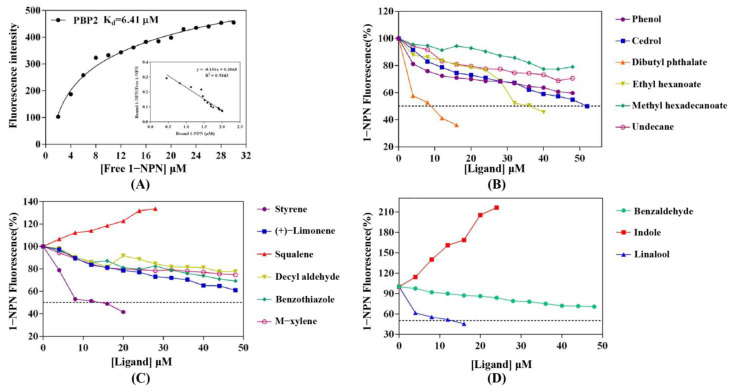
Competitive fluorescence binding assays of CbuqPBP2. (**A**) The binding curve and Scatchard plot of 1-NPN with CbuqPBP2. (**B**,**C**) Competitive binding curves of CbuqPBP2 to volatile compounds of *C. buqueti*. (**D**) Competitive binding curves of CbuqPBP2 to volatile compounds of *N. affinis*.

**Figure 5 ijms-24-16925-f005:**
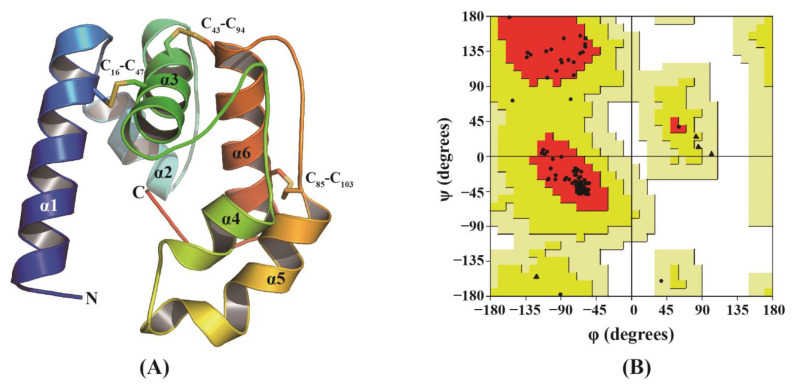
(**A**) Three-dimensional (3D) structural model of CbuqPBP2. Disulfide bonds are indicated by C_16_-C_47_, C_43_-C_94_, and C_85_-C_103_. (**B**) Ramachandran plot analysis.

**Figure 6 ijms-24-16925-f006:**
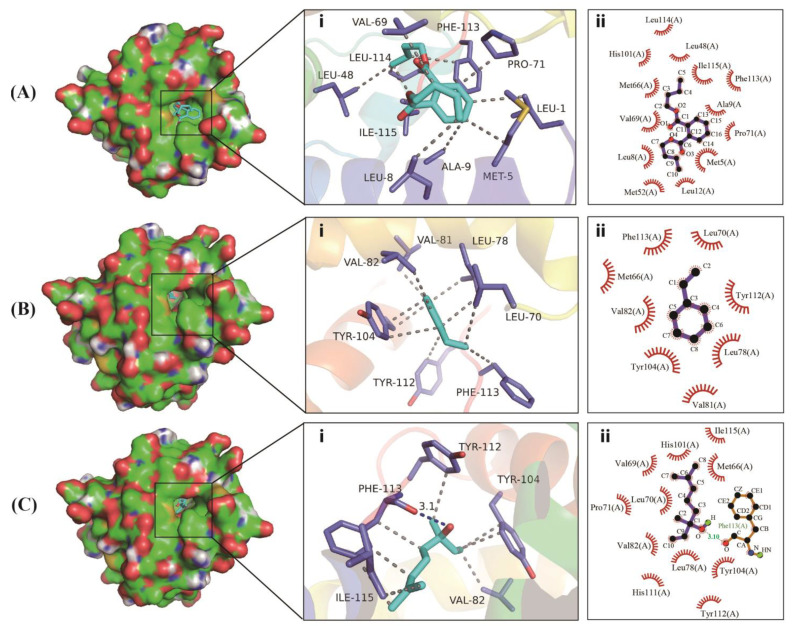
Binding modes of CbuqPBP2 with dibutyl phthalate (**A**), styrene (**B**), and linalool (**C**). (**i**) Three-dimensional demonstrations of the binding interface. (**ii**) Two-dimensional demonstrations of the detailed binding of the key residues with volatile compounds.

**Figure 7 ijms-24-16925-f007:**
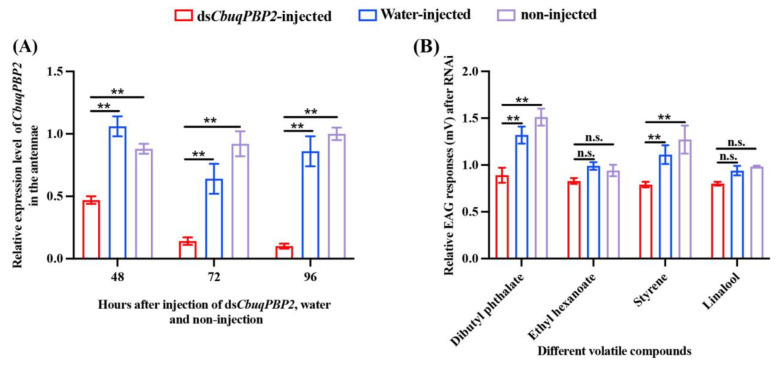
RNA interference of *CbuqPBP2*. (**A**) Relative expression levels of *CbuqPBP2* in the antennae of ds*CbuqPBP2*-injected, water-injected, and non-injected male adults after 48, 72, and 96 h. (**B**) Relative EAG responses of male adults to four volatile compounds after the knockdown of *CbuqPBP2* via RNAi. Statistically significant differences between dsRNA injection and control groups are indicated by asterisks (*p <* 0.01). The n.s. indicates that the difference is not significant.

**Table 1 ijms-24-16925-t001:** Binding affinities of CbuqPBP2 with *C. buqueti* and host plant volatiles.

Ligand	Structural Formula	CAS No.	IC_50_ (μM)	*K*_i_ (μM)
*C. buqueti* volatiles
Phenol	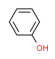	108-95-2	― *^a^*	―
Cedrol	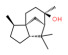	77-53-2	―	―
Dibutyl phthalate	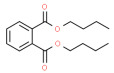	84-74-2	7.30	6.32
Ethyl hexanoate	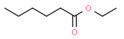	123-66-0	45.32	39.20
Methyl hexadecanoate	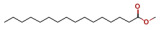	112-39-0	―	―
Undecane	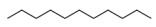	1120-21-4	―	―
Styrene	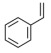	100-42-5	13.14	11.37
(+)-Limonene	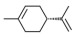	5989-27-5	―	―
Squalene	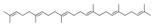	111-02-4	―	―
Decyl aldehyde	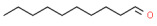	112-31-2	―	―
Benzothiazole	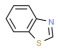	95-16-9	―	―
M-xylene	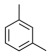	108-38-3	―	―
*N. affinis* volatiles
Benzaldehyde	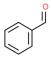	100-52-7	―	―
Indole	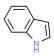	120-72-9	―	―
Linalool	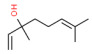	78-70-6	12.20	10.55

*^a^* It was considered that the protein had no binding with the tested ligands if the IC_50_ values > 50 μM and *K*_i_ values were not calculated, and these two values are represented as “―”.

## Data Availability

Data are contained within the article and Appendix A.

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
