# Peer review of "Molecular and Functional Characterization of Pheromone Binding Protein 2 from Cyrtotrachelus buqueti (Coleoptera: Curculionidae)"

_ijms, 2023, doi:10.3390/ijms242316925_

Round 1

Reviewer 1 Report

Comments and Suggestions for Authors

The manuscript describes work that has been undertaken to characterise the binding characteristics of several ligands to the PBP CbuqPBP2.  A range of complementary in silico and in vitro approaches have been combined to explore these interactions.  The results are of interesting in that they demonstrate an efficient workflow for probing OBP-ligand interactions.  However, I have serious concerns as to the rational behind the ligands selected for this study with no relevant ecological role provided for the compounds selected (e.g. why where 12 compounds and only 3 host plant volatiles chosen?).  Several of the compounds included and highlighted in this study include known man-made plasticizers that have never been reported from natural sources and are common contaminants from laboratory equipment.  To suggest that these compounds may be the sex pheromone of this beetle is spurious at best, and requires much stronger further evidence.  Furthermore, the references provided in the manuscript to rationalise this choice of compounds is similarly lacking evidence of any behavioural data.  While it is interesting to note that dibutyl phthalate elicits an EAG response, it is not possible to determine any potential behavioural activity of this compound without performing such experiments.

In addition, there is no rational provided as to why this particular PBP was chosen out of all of the other OBPs encoded for in the beetle.  There needs to be a much clearer rational as to why this protein was selected.

While I do believe the methodology presented here is of merit and thorough, I have strong reservations as to the relevance of the results given the poor rational of ligand selection.  I would strongly encourage the authors to provide a clearer rational as to why these compounds were chosen from an ecological stand point.  Furthermore, if the authors wish to claim that certain compounds may constitute a pheromonal component, then they must provide supportive accompanying behavioural data.

Line 56-58: The statement that the main function of OBPs is transportation of sex pheromones is not fully correct, as there is substantial evidence that they also act as a pre-screen of olfactory stimuli.

Line 95: Why would there be transmembrane domains in a protein the authors have already stated is from a family of soluble proteins.

Line 108: The authors must be consistent with their abbreviations.  The pheromone binding protein abbreviation changes between PBP and PhBP throughout the manuscript.

Axis labels: The axis labels in Figure 3 and 7 are lacking clarity as to what exactly is being reported.

Line 152: Why is only a estimate of the protein concentration being used?  It is vitally important for fluorescence binding assays that the exact concentration of protein is known to provide accurate titration data.

Line 190: The authors should clarify in the main body of text that AutoDock was used for the docking studies.  I note that the AutoDock reference is missing from the references.

Line 265-266: The comment that a common man-made contaminant is potentially a sex pheromone component is completely unfounded given the evidence that has been presented here. There is no behavioural data to support this assertion at all.  Just because a compound binds to an olfactory protein and elicits an EAG response does not mean that it will be behaviourally active.

Comments on the Quality of English Language

I believe that the quality of the English needs to be throughout the manuscript. The sentence structure throughout the manuscript needs to be improved to provide information in a timely and efficient manner.

Reviewer 2 Report

Comments and Suggestions for Authors

The paper describes the molecular and functional characterization of the pheromone binding protein 2 from the insect C. buqueti. This protein family plays key roles in insects, notably in the binding and transport of sex pheromones.

The results are clear and described in sufficient details.

I would, however, add a few points for discussion to complete the manuscript:

- In general, the comparison with CbuqPBP1 should be more systematic: in fig1 for the sequence alignment, in the tissue expression part, in the binding assay, in the discussion. 

- The results of the binding assay should be a little bit more described. In particular explain the case of squalene and indole.

- you should comment the choice of the compounds you are using for the docking studies. Why for ex. ethyl hexanoate?

Some minor points:

- for the yield of protein purification, the volume of culture and the quantity of protein in mg should be mentioned, rather than the concentration (mg/ml).

- The docking poses (Fig.6) could be zoomed in for a better view.

Round 2

Reviewer 1 Report

Comments and Suggestions for Authors

I am satisfied that the authors have addressed my main concerns originally reported.

Comments on the Quality of English Language

The authors have made good progress in correcting the English language.

Author Response

Dear Reviewer,

       We are writing to express our gratitude for your valuable feedback on the manuscript. We are delighted to learn that you are satisfied with the revised version. Thank you for your time and effort in reviewing our article, and we are indebted to you for your guidance and support.

Best regards,

Dr. Long Liu

Dr. Hua Yang, Corresponding author